# Pure Ductal Carcinoma In Situ of the Breast: Analysis of 270 Consecutive Patients Treated in a 9-Year Period

**DOI:** 10.3390/cancers13030431

**Published:** 2021-01-23

**Authors:** Corrado Chiappa, Alice Bonetti, Giulio Jad Jaber, Valentina De Berardinis, Veronica Bianchi, Francesca Rovera

**Affiliations:** SSD Breast Unit–ASST-Settelaghi Varese, Senology Research Center, Department of Medicine, University of Insubria, 21100 Varese, Italy; alicebonetti.1@gmail.com (A.B.); gjjaber@studenti.uninsubria.it (G.J.J.); valentina.deberardinis@asst-settelaghi.it (V.D.B.); veronica.bianchi@asst-settelaghi.it (V.B.); francesca.rovera@uninsubria.it (F.R.)

**Keywords:** DCIS, ductal carcinoma, surgery, breast cancer

## Abstract

**Simple Summary:**

Ductal carcinoma in situ (DCIS) accounts for 20 to 25% of all breast cancers and its incidence of progression to invasive ductal carcinoma is at least 13 to 50%. The aim of our retrospective observational analysis is to review the issues of this histological type of cancer. We confirmed in a wide population of 270 consecutive patients who underwent surgery in a single institute that the management of DCIS can be difficult and particularly complex. There are many variables to be taken into consideration such as the choice of the diagnostic and bioptical technique. This delicate management must be carried out in specialized centres such as Breast Units involving multiple professional figures to define and guarantee the best possible treatment for each patient.

**Abstract:**

Introduction: Ductal carcinoma in situ (DCIS) is an intraductal neoplastic proliferation of epithelial cells that are confined within the basement membrane of the breast ductal system. This retrospective observational analysis aims at reviewing the issues of this histological type of cancer. Materials and methods: Patients treated for DCIS between 1 January 2009 and 31 December 2018 were identified from a retrospective database. The patients were divided into two groups of 5 years each, the first group including patients treated from 2009 to 2013, and the second group including patients treated from 2014 to 2018. Once the database was completed, we performed a statistical analysis to see if there were significant differences among the 2 periods. Statistical analyses were performed using GraphPad Prism software for Windows, and the level of significance was set at *p* < 0.05. Results: 3586 female patients were treated for breast cancer over the 9-year study period (1469 patients from 2009 to 2013 and 2117 from 2014 to 2018), of which 270 (7.53%) had pure DCIS in the final pathology. The median age of diagnosis was 59-year-old (range 36–86). In the first period, 81 (5.5%) women out of 1469 had DCIS in the final pathology, in the second, 189 (8.9%) out of 2117 had DCIS in the final pathology with a statistically significant increase (*p* = 0.0001). From 2009 to 2013, only 38 (46.9%) were in stage 0 (correct DCIS diagnosis) while in the second period, 125 (66.1%) were included in this stage. The number of patients included in clinical stage 0 increased significantly (*p* = 0.004). In the first period, 48 (59.3%) specimen margins were at a greater or equal distance than 2 mm (negative margins), between 2014 and 2018; 137 (72.5%) had negative margins. Between 2014 and 2018 the number of DCIS patients with positive margins decreased significantly (*p* = 0.02) compared to the first period examined. The mastectomies number increased significantly (*p* = 0.008) between the 2 periods, while the sentinel lymph node biopsy (SLNB) numbers had no differences (*p* = 0.29). For both periods analysed all the 253 patients who underwent the follow up are currently living and free of disease. We have conventionally excluded the 17 patients whose data were lost. Conclusion: The choice of the newest imaging techniques and the most suitable biopsy method allows a better pre-operative diagnosis of the DCIS. Surgical treatment must be targeted to the patient and a multidisciplinary approach discussed in the Breast Unit centres.

## 1. Introduction

Ductal carcinoma in situ (DCIS) is an intraductal neoplastic proliferation of epithelial cells which are confined within the basement membrane of the breast ductal system [1]. DCIS is defined “pure DCIS” when it is without any invasive components [2]. It is a non-obligate precursor of the invasive ductal carcinoma (IDC). There is a wide variation in the natural history of DCIS with an estimated incidence of progression to invasive ductal carcinoma being at least 13 to 50% [3] but figuring out which DCIS will progress into an IDC is still an issue [4]. DCIS at present accounts for 20 to 25% of all breast cancers [2,5]. Since the introduction of mammography in breast cancer screening, DCIS incidence increased consistently. The most common identification of DCIS is through the detection of atypical micro-calcifications on screening mammography [2]. Like all breast cancers, three sequential and precise steps are important for DCIS diagnosis: clinical examination, imaging (bilateral mammography and bilateral breast ultrasound and in few selected cases magnetic resonance imaging) and biopsy. Today, the majority of DCIS diagnoses are made preclinical with mammography, with 75% of the cases presenting as microcalcifications, which are a result of cellular necrosis. Microcalcifications are seen on many mammograms and there are well-described patterns that help to distinguish benign from potentially malignant changes [6,7]. Breast ultrasound is useful in further characterizing the suspicious lesions or calcifications detected on screening mammography and for image-guided biopsy [6]. Magnetic resonance imaging (MRI) is a second-level diagnostic survey, supplementing mammography and ultrasound. Presentation of pure DCIS is more variable on MRI compared to invasive carcinoma. MRI has a sensitivity ranging from 70 to 100% for identification of DCIS [8,9,10,11,12]; this sensitivity increased for a high histologic grade [10,11,12,13]. Biopsy and histological examination are necessary to confirm the imaging suspect of DCIS. Biopsy has a sensitivity of 97.3% and specificity of 99.7% for detection of carcinoma. For the detection of DCIS, sensitivity is 65.4% and specificity is 97.7%. The lower sensitivity, in this case, is a result of the discovery of invasive carcinoma on the subsequent excision specimen [14]. If lesions are visible on ultrasonography, this is the preferred method [15,16,17,18,19,20]. Vacuum-assisted biopsy represents an additional biopsy method, which results in a lower rate underestimation of lesions due to its ability to obtain more tissue.

Local and systemic treatment of DCIS aims to prevent the onset of invasive carcinoma [21] after performing a complete surgical excision. There is considerable controversy over margins resections extension in DCIS. Sentinel lymph node biopsy is not indicated in conservative surgery regardless of grading but can be indicated only in the presence of multiple clusters of microcalcifications, and extensive lesion which requires a mastectomy or in patients in whom surgical treatment may compromise the subsequent sentinel lymph node procedure [22,23]. Currently, the therapeutic standard of DCIS is conservative surgery with negative margins associated with postoperative radiotherapy [24]. Now, a debate on radiotherapy benefits after conservative surgery in DCIS treatment is open [25,26,27].

The purpose of this work is to retrospectively evaluate our experience in the treatment of pure DCIS cases. The secondary purposes are to analyse the anatomical pathological features, the type of surgery, the involvement of resection margins, the percentage of surgical radicalization and the adjuvant therapy performed. All the data obtained in this review are in line with those reported in the most recent literature.

## 2. Materials and Methods

Using the data contained in the database of the SSD Breast Unit of ASST-Sette Laghi—University of Insubria of Varese, we identified among the 3586 patients treated consecutively for breast cancer which ones had a diagnosis of “pure” DCIS in the period between 1 January 2009 and 31 December 2018. We performed a retrospective observational study that analyses 270 patients with “pure” DCIS diagnosis who underwent surgery in the period examined. All the patients examined for our study are female. The patients were divided into two groups of 5 years each, the first including patients treated from 2009 to 2013, and the second patients treated from 2014 to 2018.

The number of patients treated in the first period (2009–2013) was 81 while the number treated in the second period (2014–2018) was 189. Of these patients the following data was collected and subsequently entered into a database:-Age at surgery-Menopausal state-Familiarity-Preoperative stage-Type of surgery-Sentinel lymph node biopsy (SLNB)-Radicalization-Dimensions-Comedonecrosis-Paget’s disease-Specimen margins-Grading-Hormonal receptors for estrogens and progesterone-Adjuvant therapy

A preoperative histological examination led to three possible clinical stages: stage 0, stage IA, and stage IIA. In some patients, it was not possible to define a clinical-stage (stage X). At the postoperative histological examination, all the tumours resulted as pure DCIS and therefore returned to stage 0. The surgical approach can be divided into 2 categories: conservative surgery (lumpectomy, nodulectomy and excisional biopsy) and radical surgery (mastectomy). Taking into account the tumour’s size, patients were divided into two groups. According to the date reported in the literature we decide for a 2 cm cut off as done in the work Co et al. [28]. In the first group are neoplasms with dimensions greater than or equal to 2 cm while in the second group are neoplasms less than 2 cm. The margins were considered positive if they were less than 2 mm; and negative when the margins were greater than or equal to 2 mm. There were 4 histological grades found for our patients: Grade X—not determined, grade G1—well-differentiated, grade G2—moderately differentiated and grade G3—poorly differentiated.

Postoperative treatments offered to our patients can be grouped into 4 groups:(1)patients undergoing both therapies: radiotherapy + hormone therapy(2)patients subjected to radiotherapy only(3)patients subjected to hormone therapy only(4)patients not subjected to any adjuvant treatment (surgical follow-up)

A statistical analysis was performed to see significant differences in the 2 periods taken into consideration (2009–2013 and 2014–2018). We compared the data obtained from our database with those in the literature. Follow-up data were collected until 5 July 2019. The significance analysis of the parameters was evaluated with Fisher’s exact test. The level of significance was established at *p* < 0.05. Statistical analyses were performed using the GraphPad Prism software for Windows.

## 3. Results

From 1 January 2009 to 31 December 2018, 3586 patients underwent surgery for breast cancer at the SSD Breast Unit of ASST Sette Laghi; of these, 270 had a definitive diagnosis of “pure” DCIS at the histological examination. The patients analyzed were divided into two periods, each of 5 years. We have decided to divide the patients into two distinct groups because in our centre since 2014 we have introduced some intraoperative strategies to improve the accuracy and precision of surgical resection. Since 2014 we have routinely performed the intraoperative assessment of the surgical resection margins performed by the pathologist who “in real time” guides the surgeon to a possible widening of the surgical excision margins. Another strategy, in the case of microcalcifications, was to perform an x-ray of the specimen to evaluate their complete excision. In our institute, since 2014, few surgeons have been dedicated exclusively to the Breast Unit, which has allowed the development of an ultra-specialization with an improvement in intra and post-operative results. The period between 2009 and 2013 will be called “the first period” while the one between 2014 and 2018 will be called “second period”. In the first period 1469 patients were operated, of these 81 (5.5%) with DCIS, while in the second period 2117 of these 189 (8.9%) with DCIS. The incidence of DCIS was therefore significantly increased between the first and second period (*p* = 0.0001). The age at surgery was between 36 and 86 years. In the first period the average age at surgery was 59.7 years (±12 years), and of these 56 (69%) were postmenopausal, in the second period the average age was 59.8 years (±10.9 years) and of these 137 (72.5%) were postmenopausal. We can conclude that there were no differences in the age of onset of the disease and the menopausal state between the 2 groups. Familiar history was positive for breast cancer in 24 (29.6%) patients in the first period, while between 2014 and 2018 it was positive in 69 (36.5%) patients (*p* = 0.33). One hundred sixty-three (60.7%) patients were in stage 0 at the breast cancer diagnosis and therefore had a correct diagnosis of “pure” DCIS. In the first period, 38 (46.9%) patients were included in this group, in the second period 125 (66.1%) were in this group. The other patients were included in three other possible classes. In the first period, 25 (30.9%) patients were considered in stage IA, 4 (4.9%) patients in stage IIA and 14 (17.3%) in stage X (no possibility to define clinical stage). Between 2014 and 2018, 19 (10.1%) in stage IA, 4 (2.1%) in stage IIA and 41 (21.7%) in stage X (Table 1).

The number of patients included in clinical stage 0 increased significantly (*p* = 0.004) between the first and second period, while those in stage IA decreased significantly (*p* < 0.0001). For the other two stages, there were no statistically significant differences. We can conclude that between the first and the second period the number of patients who had a correct clinical staging increased.

All patients were surgically treated. Conservative surgery was performed in 193 (71.5%) patients, while the other 77 (28.5%) underwent mastectomies. Between 2009 and 2013, conservative surgical treatment was performed on 67 (82.7%) patients. Lumpectomy were 52 (64.2%), nodulectomies 4 (5%) and excisional biopsies 11 (13.6%). The other 14 (17.3%) patients were surgically treated with mastectomy. In the second period, out of 189 patients, 126 (66.7%) had conservative treatment. Of these, 107 (56.6%) had a lumpectomy, 2 (1.1%) had a nodulectomy and 17 (9.4%) had an excisional biopsy. The other 63 (33.3%) patients were treated with mastectomy. In the first period 48 (59.3%) of the patients also underwent a sentinel lymph node biopsy (SLNB) while in the second 98 were (51.9%) (Table 2).

Between the first and the second periods, the number of patients who underwent radical treatment increased significantly (*p* = 0.008). Regarding the other types of surgical treatments, no statistically significant differences were observed between the 2 periods examined. SLNB numbers overlapped over time (*p* = 0.29).

DCIS patients were divided into 2 groups, those with dimensions greater than or equal to 2 cm and those smaller than 2 cm. Between 2009 and 2013, patients with tumours with greater dimensions than or equal to 2 cm were 14 (17.3%) while those with tumours with smaller dimensions than 2 cm were 64 (79.0%). In this first period, it was not possible to find the dimensions in 3 (3.7%) cases. Between 2014 and 2018 there were 26 (13.8%) DCIS with greater dimensions than or equal to 2 cm. In 161 (85.2%) cases the neoplasm size was less than 2 cm. In 2 (1.1%) cases we did not have the tumour size (Table 3).

Surgical margins are an important aspect to consider when discussing DCIS. Recent guidelines consider positive margins when the tumour is less than 2 mm from the margins of resection. If the distance is 2 mm or greater, the margins are considered negative. In the first period, 48 (59.3%) had negative margins, in 25 (30.9%) the neoplastic tissue reached the margins at a distance less than 2 mm and in 8 cases (9.9%), it was not possible to determine the distance to the margins of resection. Of the 25 patients with positive margins, 16 (64%) received radiotherapy in combination with hormone therapy as adjuvant therapy, 3 (12%) received only radiotherapy, 4 (16%) received only endocrine therapy and 2 (8%) cases were decided for surgical radicalization. Radiotherapy was performed on a total of 19 (76%) patients. Between 2014 and 2018, 137 (72.5%) DCIS patients were at a greater or equal distance than 2 mm, 34 (18.0%) had positive margins and in 18 cases (9.5%) there was no clarification on the margins. Of the 34 patients with positive margins, 14 (41.2%) received radiotherapy in combination with hormone therapy as adjuvant therapy, 11 (32.4%) patients with radiotherapy alone, 5 (14.7%) only endocrine therapy, 3 (8.8%) patients underwent surgical re-excision and 1 patient (2.9%) received no adjuvant therapy. Radiotherapy was therefore performed on a total of 25 (73.5%) patients (Table 4).

In the 2 periods analysed, there was a significant difference between the number of tumours with positive margins and the number of tumours with negative margins. Indeed in the period between 2014 and 2018, the number of DCIS cases with positive margins decreased significantly (*p* = 0.02) compared to the first period examined, and consequently the number of interventions with negative margins increased significantly (*p* = 0.045).

There were 4 possible histological grades for postoperative histological examinations. In the period between 2009 and 2013, 13 (16%) DCIS cases were well-differentiated (grade G1), 39 (48.2%) were grade G2, 23 (28.4%) were grade G3. In 6 (7.4%) patients it was not possible to find the histological grade. In the second period examined we had instead 46 (24.3%) grade G1, 73 (38.6%) grade G2, 64 (33.9%) grade G3 and in 6 (3.3%) cases it was not possible to determine the histological degree. No significant differences were found in the 2 periods examined (Table 5).

In 85.6% of the cases there was positivity for estrogen receptors (ER+); for progesterone receptors (PgR) positivity was 79.3%. In the first period, 69 (85.2%) patients were positive for both estrogen and progesterone receptors. In the second period, 162 (85.7%) patients had estrogen receptors and 145 (76.7%) had progesterone hormones. No statistically significant differences were observed in the positivity of hormone receptors (Table 6).

For adjuvant therapy, patients can be divided into 4 groups. A group of patients subjected to radiotherapy and hormone therapy, a group subjected only to radiotherapy, another group subjected only to hormone therapy and in the last group of patients did not receive any adjuvant treatment. The guidelines set radiotherapy as a gold standard for adjuvant therapy to which hormone therapy can be added if estrogen and progesterone receptors are positive [29]. The hormonal therapy is also chosen based on the patient’s menopausal status and the two most used drugs were tamoxifen for pre-menopausal or aromatase inhibitor for post-menopausal patients. The duration of hormone therapy was 5 years. A standard therapeutic protocol of radiotherapy with a whole breast irradiation (WBI) typically to a dose of 50 Gy in 25 fractions and a tumour bed boost of 10 Gy in 5 fractions is always performed for patients undergoing conservative surgical treatment. In the first period, 39 (48.2%) patients were included to the group in which adjuvant therapy was composed of radiation therapy plus hormone therapy, 13 (16.1%) patients returned to the group to which only radiotherapy was administered, 15 (18.5%) in the group treated with hormone therapy only, and in the no adjuvant therapy group there were 14 (17.3%) patients. In the second period, 79 (41.8%) patients received as adjuvant therapy radiotherapy in combination with hormone therapy, 29 (15.3%) patients were treated only with radiotherapy, 44 (23.3%) with hormone therapy, and 37 (19.6%) patients had no adjuvant treatment (Table 7).

No significant differences were found in the 2 periods examined. Adjuvant therapy offered to our patients remained unchanged.

All patients were followed after surgery at the SSD Breast Unit through clinical and instrumental assessments; the follow-up data were updated until 5 July 2019. In the first period, the follow up had an average of 63.5 months. We observed 3 (3.7%) local recurrences among the 81 patients undergoing conservative surgery. These relapses were found at 47.3 months after surgery. In all 3 cases. patients had received radiotherapy as adjuvant therapy and in 2 of these the hormone therapy had been added. At the definitive histological examination, the margins were positive in 2 cases associated with a poorly differentiated histological grade G3 and 1 case presented positive margins with a moderately differentiated histological grade G2. In 8 (9.9%) cases the data at follow up were missing and/or incomplete. In the second period the follow up lasted on average 28.9 months. 2 (1.1%) recurrences were observed at 26 months after conservative surgery. Of these two cases: 1 patient had positive margins and a G3 histological grade and underwent adjuvant radiation therapy while the other patient had negative margins and a G2 histological grade and had not undergone any adjuvant treatment. There were 9 patients whose data were missing and/or incomplete at follow-up (5.3%).

For both periods analysed all 253 patients who were followed are currently living and free of disease; we have conventionally excluded the 17 patients whose data were lost.

## 4. Discussion

During the 10 years (2009–2018) evaluated in our study, pure DCIS represented 7.5% of all breast carcinomas treated consecutively at the SSD Breast Unit—ASST-Sette Laghi of Varese. Eighty-one out of 1469 (5.5%) between 2009 and 2013 and 189 out of 2117 (8.9%) between 2014 and 2018 were pure DCIS cases. A statistically significant increase (*p* = 0.0001) was therefore observed in the incidence of this breast cancer type. DCIS incidence reported in the literature is approximately 20% [27,30,31], this is higher than the incidence of our centre. However, it must be considered, that DCIS is presented in a non-negligible percentage of cases associated with an infiltrating component, and these may have been included in the percentages of the cited studies. For example, in the study by Park et al. [32] of 201 patients observed retrospectively, at the definitive histological examination, 76 (37.8%) had an infiltrating DCIS diagnosis. In Co et al. Study [28], pure DCIS incidence was instead equal to 6.7%. Conflicting results in the literature on the exact incidence of “pure” DCIS are present but there is a common statement that the incidence is increasing. This was obtained with the introduction of the screening program started around the mid-1980s. DCIS is diagnosed at histological examination following suspicious mammographic images. The introduction of new imaging techniques such as digital mammography and tomosynthesis has allowed obtaining more and more detailed images with further refinement of the diagnostics, especially for those patients with a high breast density. The launch of the national mammography screening, the evolution of diagnostic imaging and the increase in awareness campaigns that led to greater attention to prevention by patients, were responsible for increasing the incidence of “pure” DCIS observed also at the SSD Breast Unit of ASST—Sette Laghi. DCIS represents stage 0. However, it must be considered that in certain cases the postoperative histological examination does not confirm the histological examination carried out before the surgery.

Some DCIS can be described as invasive carcinomas at the pre-operative histological examination and consequently treated [33]. Others are defined preoperatively as precancerous lesions, for example, atypical ductal hyperplasia, but at the pathological staging, they will be DCIS. This occurs in a non-negligible percentage of cases, from 13% to 31% [34,35,36,37,38,39,40,41,42,43] (Table 8).

Atypical ductal hyperplasia has cells looking like low-grade DCIS cells and also has cells without atypia proliferation [44]. In some cases it is not easy to distinguish between atypical ductal hyperplasia and DCIS, we are often facing borderline lesions. At the histological level, criteria have been put in place to help the anatomopathologist in defining the lesion. If the lesion exceeds 2 mm or if we have the involvement of 2 or more mammary ducts we will talk about DCIS, otherwise, we talk about atypical ductal hyperplasia [45,46]. These are sufficiently precise criteria that make it possible to differentiate DCIS from atypical ductal hyperplasia but it is not always possible at a histological level to be sure, and consequently, the clinical stage can be under/overestimated [47]. Benign histological results on surgical specimen final anatomopathological evaluation can be found because the biopsy completely remove the DCIS in almost 2–28% of the cases [47,48,49,50,51] (Table 9).

DCIS underestimation in the preoperative histological examination highlights the assignment criticism to a correct clinical-stage, conditioning consequently a different therapeutic approach. In fact, only 163 (60.7%) out of 270 “pure DCIS” confirmed at the postoperative histological examination were also found to be DCIS at the clinical stage. In 52 (19.3%) cases the preoperative histological examination was compatible with infiltrating carcinomas (stage IA or stage IIA) and in the remaining 55 (19.3%) cases, patients were assigned to stage X, including atypical ductal hyperplasia and cases in which histology failed to precisely define the stage. These percentages are similar to those observed in the literature. Between the first period (2009–2013) and the second period (2014–2018), we witnessed a statistically significant increase (*p* = 0.004) of patients returning to a clinical-stage 0. This means that preoperative histology was found to be more precise. This may have been caused by the increased use of vacuum-assisted biopsy (VAB). VAB allows a better preoperative histological examination but does not reset the possibility of having an upgrade with the surgical sample. Several studies show that with VAB the possibility of having an underestimation of the lesion on the biopsy is halved [18,51,52].

Once the DCIS has been diagnosed, a therapeutic path must be taken. DCIS therapy is still under discussion [53]. Until the early 1990s, mastectomy represented the gold standard for the DCIS treatment, allowing a disease-free survival of 99.1% at 10 years and with an overall survival rate of 99.4% at 10 years [30]. Many randomized clinical trials have demonstrated the equivalence between mastectomy and conservative surgery for invasive breast cancer in the early stages [54,55,56,57,58]. Consequently, towards the end of the 1980s, this type of surgery was also adopted for DCIS although prospective randomized studies have not been done [53]. Current guidelines suggest as a standard treatment for DCIS conservative surgery with negative margins (≥2 mm), followed by radiotherapy [21]. In our study, the number of patients who underwent conservative treatment were 193 (71.5%) while those who received demolitive treatment were 77 (28.5%). These results are in line with those reported in the literature [59,60,61,62] (Table 10).

We observed a statistically significant increase in the number of mastectomies (*p* = 0.008) between the 2 periods examined. This type of treatment still indicated if the disease is too extensive to be resected conservatively with a good aesthetic result, if reaching negative resection margins is impossible, or in the case of contraindications to postoperative radiotherapy [21]. The increase number of mastectomies in the period 2014–2018 is certainly due to the improvement of diagnostic techniques with the introduction of tomosynthesis and the increasing use of MRI which have made it possible to evaluate more accurately the extent of the neoplasia and in many cases resulting in a change of the surgical strategy. Some patients decide for mastectomy for fear of relapse after conservative treatment or for fear of radiation therapy [63]. Based on these elements, mastectomy is a valid treatment for the DCIS.

In our study, margins equal to or greater than 2 mm at the definitive histological examination were considered negative. Negative margins were obtained in 179 (66.3%) cases out of 270 and positive in 57 (21.1%). In the second period that was examined the number of negative margins increased significantly (*p* = 0.045). There is considerable controversy over the necessary breadth of resection margins in carcinoma in situ and over time the definition of negative margins has changed [21]. The 2009 AIOM guidelines considered margins to be negative if they exceeded 1 mm [64]. In 2016 the “Society of Surgical Oncology”—SSO, the “American Society for Radiation Oncology”—ASTRO and the “American Society of Clinical Oncology”—ASCO published a joint Consensus Guideline (SSO-ASTRO-ASCO) about this topic. The conclusion of this study based on a systematic review of 20 studies including 7883 patients, adopted the use of 2 mm from the margin as an adequate standard for DCIS with adjuvant radiotherapy [65]. These guidelines have therefore led as a goal during surgery to have margins free from neoplastic cells greater than or equal to 2 mm and thus explains the statistically significant increase in our study.

When margins are less than 2 mm the possibility of surgical re-operation must be evaluated. In the 9 years of our study, in 5 (8.5%) of the 59 cases with positive margins, radicalization was the choice, two of these between 2009 and 2013 and the other 3 between 2014 and 2018. There was no statistically significant difference in the re-operation rates in the 2 periods. The decision to undertake a new surgical procedure must be taken after an adequate multidisciplinary evaluation, considering particularly the anatomopathological variables and the instrumental investigations performed, integrated with the clinical evaluation of the patient. If the margin is positive, radicalization is not always the only right option. In fact in literature it is observed that between 30 and 65% of mastectomies performed after conservative surgery are free of neoplastic cells [66].

Lesion localization represents a problem for the correct choice of DCIS surgery. DCIS appears in most cases as a non-palpable lesion. This leads to a more difficult intraoperative localization of the neoplastic tissue and consequently a different intra-operative management of the margins excision adequacy. In our study, the non-palpable lesions were localized using Technetium 99 m complexed with macromolecules of human albumin (radioguided occult lesion localization—ROLL) which allows a better intraoperative identification [67]. In all cases an anatomopathological intraoperative evaluation of the surgical specimen was performed to provide a first real time macroscopical indication of the margins status; in the presence of microcalcifications, the radiograph of the surgical specimen was always performed.

Sentinel lymph node biopsy (SLNB) is not indicated for DCIS, but may be indicated in presence of multiple clusters of microcalcifications, extensive lesions that require a mastectomy or in patients in whom surgical treatment may compromise the subsequent lymph node biopsy procedure [68]. In our study, the SLNB was performed in 146 (54.3%) cases without observing statistically significant differences in the 2 periods considered. The number of patients to whom SLNB is performed is in line with data found in the literature [69,70,71,72] (Table 11).

Surgical treatment is usually followed by adjuvant therapy. This can be represented by radiotherapy, hormone therapy or both. Both have been shown to reduce the number of local recurrences even if they do not affect patient survival [73]. The radiotherapy utility was demonstrated by the study NSABP B-17 [24] and by the study carried out by the ECOG-ACRIN group (The Eastern Cooperative Oncology Group—American College of Radiology Imaging Network, better known as the Eastern Cooperative Oncology Group) E5194 which carried out a prospective, non-randomized study dividing patients with DCIS based on tumour grade and size. The first group included patients with low-grade or intermediate DCIS with tumour size up to 2.5 cm. In the second group were included patients with high-grade DCIS with tumour size dimensions up to 1 cm. In the 2 groups, only conservative surgery was performed with resection margins equal to or greater than 3 mm. At 5 years the recurrence rate in the first group was 6.1% whereas in high-grade patients up to 1 cm it was 15.3% (*p* = 0.024) [74]. At 12 years, the recurrence rate in the first group was 14.4% whereas in the second group it was 24.6% (*p* = 0.003) [75]. This demonstrated the important role of radiotherapy as an adjuvant therapy, and given the statistically significant difference in the recurrence rate of the 2 groups, the DCIS histological grade seems to play an important role as an independent risk factor for recurrence. The benefit of hormonotherapy was demonstrated by the NSABP B-24 [76].

As reported in the literature, the patients treated at the SSD Breast Unit of the ASST-Sette Laghi received in most cases (160 patients out of 270; 59.3%) radiotherapy [77]. This explains the lower recurrence rate found in our centre compared to the ECOG-ACRIN E5194 study which demonstrated the importance of radiotherapy as an adjuvant therapy. In fact, in the first period taken into consideration in our study, 3 (3.7%) recurrences were found, 2 of which had positive margins and a poorly differentiated histological grade (G3) and 1 had positive margins with a moderately differentiated histological grade (G2). Between 2014 and 2018, 2 (1.1%) recurrences were observed, 1 patient had positive margins and a G3 histological grade while the other patient had negative margins and a G2 histological grade. These results confirm that the high histological grade DCIS is a risk factor for local recurrence as observed in the ECOG-ACRIN E5194 study. The margins positivity is added as a risk factor for local recurrence to the high histological level [78]. Surgical radicalization should be considered in these cases. During conservative surgery, the surgeon must balance the risk of local recurrence to cosmetics to not compromise the prognosis [66].

According to recent literature, we do not routinely use molecular tests because no strategy incorporating the Oncotype DX DCIS Score was cost effective. Oncotype DCIS assay became commercially available and now provides limited accessibility to cost coverage [79], there is limited experience with regard to the clinical utility of the Oncotype DX Score [80].

## 5. Conclusions

From diagnosis to therapy, difficulties can be encountered that can make the DCIS therapeutic pathway particularly complex. There are many variables to be taken into consideration and multiple professional figures must be involved to define and guarantee the best possible treatment for each patient. As we have seen, the introduction of new imaging techniques and the increasing use of MRI has allowed increase the diagnosis of “pure DCIS”. In our series the vacuum-assisted biopsy (VAB) has shown a decrease in the under/overestimation of the preoperative DCIS.

As we have seen in our study, conservative surgery with negative margins (≥2 mm) followed by radiotherapy is the treatment that in most cases is carried out, but mastectomies are still indicated for extensive disease or if adjuvant therapy are contraindicated. We observe that the introduction of anatomopathological intraoperative evaluation of surgical margins and the specimen x-ray (in case of microcalcification) are crucial to choose the right surgical option for the patient and to move from conservative surgery to mastectomy.

Diagnostic and therapeutic management of the DCIS is still an open challenge as it requires accurate planning structured and shared by a multidisciplinary team. This delicate management must be carried out in specialized centres such as the Breast Units which take care of the patient and accompany her on this delicate journey.

As we have seen in the 270 patients treated consecutively at the SSD Breast Unit of the ASST Sette Laghi, DCIS has an extremely good prognosis, and it is reported that some DCIS cases do not progress to invasive cancer even without treatment. However, presently, it is not possible to reliably identify a population that does not progress to invasive cancer even without treatment.

It is necessary to establish new less invasive treatment strategies for DCIS without compromising the good prognosis obtained with the current treatment approach. If it were possible, the DCIS treatment could be even more personalized.

## Figures and Tables

**Table 1 cancers-13-00431-t001:** Patients divided by stage.

Clinical Stage	2009–2013 *N* = 81	2014–2018 *N* = 189	2009–2018 *N* = 270	*p*-Value
0	38 (46.9%)	125 (66.1%)	163 (60.7%)	0.004
IA	25 (30.9%)	19 (101%)	44 (17%)	<0.0001
IIA	4 (4.9%)	4 (2.1%)	8 (3%)	0.25
X	14 (17.3%)	41 (21.7%)	55 (19.3%)	0.51

**Table 2 cancers-13-00431-t002:** Surgical treatments performed.

Surgery	2009–2013 *N* = 81	2014–2018 *N* = 189	2009–2018 *N* = 270	*p*-Value
Lumpectomy	52 (64.2%)	107 (56.6%)	159 (58.9%)	0.28
Nodulectomy	4 (5%)	2 (1.1%)	6 (2.2%)	0.07
Escissional Biopsy	11 (13.6%)	17 (9.4%)	28 (10.4%)	0.28
Mastectomy	14 (17.3%)	63 (34.8%)	77 (28.5%)	0.008
SLNB	48 (59.3%)	98 (51.9%)	146 (54.3%)	0.29

**Table 3 cancers-13-00431-t003:** Patients subdivision according to the tumour size.

Tumour Size	2009–2013 *N* = 81	2014–2018 *N* = 189	2009–2018 *N* = 270	*p* Value
<2 cm	64 (79.0%)	161 (85.2%)	225 (83.3%)	0.22
≥2 cm	14 (17.3%)	26 (13.8%)	40 (14.8%)	0.46
X	3 (3.7%)	2 (1.1%)	5 (1.9%)	0.16

**Table 4 cancers-13-00431-t004:** Division of patients based on resection margins.

Margins	2009–2013 *N* = 81	2014–2018 *N* = 189	2009–2018 *N* = 270	*p*-Value
Negatives (≥2 mm)	48 (59.3%)	137 (72.5%)	185 (68.5%)	0.045
Positives (<2 mm)	25 (30.9%)	34 (18.0%)	59 (21.9%)	0.02
X	8 (9.9%)	18 (9.5%)	26 (9.6%)	0.99

**Table 5 cancers-13-00431-t005:** Subdivision by histological degrees of the surgical specimens.

Grading	2009–2013 *N* = 81	2014–2018 *N* = 189	2009–2018 *N* = 270	*p*-Value
G1	13 (16%)	46 (24.3%)	59 (21.9%)	0.15
G2	39 (48.2%)	73 (38.6%)	110 (40.7%)	0.18
G3	23 (28.4%)	64 (33.9%)	87 (32.2%)	0.40
X	6 (7.4%)	6 (3.2%)	14 (5.2%)	0.19

**Table 6 cancers-13-00431-t006:** Hormone receptors positivity.

Hormone Receptors	2009–2013 *N* = 81	2014–2018 *N* = 189	2009–2018 *N* = 270	*p*-Value
ER+	69 (85.2%)	162 (85.7%)	231 (85.6%)	>0.99
PgR+	69 (85.2%)	145 (76.7%)	214 (79.3%)	0.14

**Table 7 cancers-13-00431-t007:** Adjuvant therapy offered to our patients.

Adjuvant Therapy	2009–2013 *N* = 81	2014–2018 *N* = 189	2009–2018 *N* = 270	*p*-Value
Radioterapy	13 (16.1%)	29 (15.3%)	42 (15.6%)	0.86
Hormonoterapy	15 (18.5%)	44 (23.3%)	59 (21.9%)	0.43
Radioterapy + Hormonoterapy	39 (48.2%)	79 (41.8%)	118 (43.7%)	0.35
None	14 (17.3%)	37 (19.6%)	51 (18.9%)	0.74

**Table 8 cancers-13-00431-t008:** Literature review: preoperative underestimation of ductal carcinoma in situ (DCIS).

Author	Biopsy Technique	“Underestimation” Percentage (%)	Total Number of Patients	Malignacy Correlated Factors
Deshaies et al. (2011) [34]	CNB	31.3%	422	ipsilateral breast symptoms, mammographic lesion other than microcalcifications alone, 14G core needle biopsy, papilloma co-diagnosis, severe ADH and pathologists with lower volume of ADH
Linsk et al. (2017) [35]	CNB	16.6%	151	maximum lesion size and radiographic presence of residual lesion
Chen et al. (2019) [36]	CNB	23.1%	140	older age, higher ADH percentage area and higher number of ADH foci
Eby et al. (2008) [37]	Stereotactic VAB	21.1%	123	lesion size and number of foci
Allison et al. (2011) [38]	Stereotactic VAB	20.6%	97	nuclear features bordering on intermediate nuclear grade, number of foci
Speer et al. (2018) [39]	Stereotactic VAB	19%	90	ADH lesions, combined ALH and LCIS
Batohi et al. (2019) [40]	VAB	13%	464	atypical intraductal epithelial proliferation, papilloma with atypia
Meijnen et al. (2007) [41]	CNB	26.2%	172	palpable lesion, presence of a mass on mammography and intermediate or poorly differentiated tumour grade
Han et al. (2011) [42]	CNB	26%	199	extent of abnormal microcalcification on mammography, and presence of a radiologic/palpable mass and solid type of DCIS
Diepstraten et al. (2013) [43]	Stereotactic CNB	28.7%	348	lesion size, number of cores retrieved at biopsy, presence of lobular cancerization, and microinvasion

**Table 9 cancers-13-00431-t009:** Literature review: rate of complete excision of DCIS with vacuum-assisted biopsy (VAB).

Author	Biopsy Technique	Percentage of Complete Excision of the Lesion	Total Number of Patients
Pfleiderer et al. (2008) [47]	Stereotactic VAB	4.4%	45
Burak et al. (2000) [48]	Stereotactic VAB	23%	86
Krischer et al. (2020) [49]	Stereotactic VAB	28.6%	58
Salem et al. (2009) [50]	Stereotactic VAB	2%	47
Rosenfield Darling et al. (2000) [51]	Stereotactic VAB	6%	289

**Table 10 cancers-13-00431-t010:** Literature review: conservative surgery vs. mastectomy.

Author	Conservative Surgery Percentage (%)	Mastectomies Percentage (%)	Factors Related with Mastctomies	Total Number of Patients
Baxter et al. (2004) [59]	72%	28%	Younger patients, comedo histology, tumours larger than 1 cm	25,206
Rakovitch et al. (2007) [60]	70%	30%	Multifocality, tumour size, high nuclear grade, surgeon’s practice pattern	727
Worni et al. (2015) [61]	70%	30%	bilateral mastectomy, younger age	21,080
Martínez-Pérez et al. (2017) [62]	72%	28%	DCIS extent relative to breast size, type of DCIS, age, patient choice, risk of recurrence	10,582

**Table 11 cancers-13-00431-t011:** Literature review—sentinel lymph node biopsy (SLNB) performed.

Author	SLNB Percentage (%)
Prendeville et al. (2015) [69]	62%
Van Roozendaal et al. (2016) [70]	51.8%
Heymans et al. (2017) [71]	66.7%
Holm-Rasmussen et al. (2017) [72]	54.3%

## Data Availability

The data presented in this study are available on request from the corresponding author. The data are not publicly available due to the inclusion of patient sensitive data.

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
