# Peer review of "Pure Ductal Carcinoma In Situ of the Breast: Analysis of 270 Consecutive Patients Treated in a 9-Year Period"

_cancers, 2021, doi:10.3390/cancers13030431_

Round 1

Reviewer 1 Report

Thank you for addressing these concerns and those of the other Reviewers. I find the revised version to be suitable for publication.

Reviewer 2 Report

Congratulations to the authors for the amended version of your submission! 

This version of the manuscript now merits publication after the reviewers revisions have been addressed.

This manuscript is a resubmission of an earlier submission. The following is a list of the peer review reports and author responses from that submission.

Round 1

Reviewer 1 Report

The authors provide a comprehensive retrospective analysis of breast cancer clinical presentation and diagnostics. Notably, the authors found that DCIS identification by preoperative histologic analysis may have an alarmingly high rate of inaccurate categorization as DCIS rather than invasive carcinoma or pre-malignant hyperplasias, indicating that the current early diagnostic assessment through histology is severely lacking and inappropriately informing clinical decision making in some cases.

In general, the manuscript is well written and the data is discussed well in the context of the literature. I have a few concerns that should be addressed to improve the impact of the manuscript.

  1. Not sure if this is from the import into the MDPI system, but there are a number of cases where there is inappropriate indentation resulting in hanging one-sentence paragraphs. These occur frequently in the Introduction, Materials/Methods and Results.
  2. Table 4: In the far-left column, please spell “Negatives” and “Positives” correctly.
  3. Table 5: “Grading” should be aligned in them middle of the row as the titles are done for other tables.
  4. Table 7: Same issue as Table 5: titles should be aligned consistently.
  5. Conclusion: Considering that only about 50% of DCIS are potentially leading to IDC, as the authors point out, it would be important for them to expand on what metrics are currently being investigated as potential biomarkers to stratify patients by risk of recurrence or metastasis, such as those described some time ago by Lari and Kuerer (J Cancer 2011). Would precision medicine in these patients be achievable through these, or possibly by incorporating Oncotype DX or other diagnostic assays? While the current manuscript is very useful, the authors stop short of asserting any potential routes towards improving patient diagnostics in light of the current findings.
  6. The authors state that Chi-square was used for these analyses. However, they use a relatively small number of samples compared to those generally indicated for Chi-square. Data should be re-examined by Fisher’s Exact Test for increased stringency and accuracy.

Author Response

Dear Editor,

Thank you very much for your helpful comments. We have done our best to address your concerns.

I try to answer point by point to your concerns:

1) The inappropriate indentation are due to the import of the document into the MDPI system. We check the entire manuscript in all the sections to adjust the incorrect one-sentence paragraphs.

2) In table 4 we have spelt in the correct way the words required.

3) We have aligned the word “Grading” in the middle of the row of table 5.

4) The same thing of table 5 is done also in table 7.

5) To stratify and identify patients who have a high risk of recurrence and / or metastatisation, we routinely analyze and identify some factors at the final histological examination. These factors are ER and PGR status, HER2 expressions, Ki67 and p53.

According to recent litterature we do not routinely use molecular tests because no strategy incorporating the Oncotype DX DCIS Score was cost effectiveness. Oncotype DCIS assay became commercially available and limited accessibility to cost coverage, there is limited experience with regard to the clinical utility of the Oncotype DX Score.

6) We analyzed our data using the Fischer's Exact Test and we obtained statistically significant results comparable to the Chi-square test with the only exception of the noduletomies that are no longer significant. We modified the text by inserting the new data obtained through the use of the Fischer Exact Test.

Thanking you once more for your useful comments, we sincerely hope that you will find the revisions satisfactory.

Reviewer 2 Report

The authors obtained data set contain 270 patients from database (SSD Breast Unit of ASST-Sette Laghi). And then the authors divided the patients into two groups, and compared the two groups from many different angles like clinical stage, surgery, and so on. The authors did get some interesting results. However, the novelty and the contribution to the community is not enough. 

Author Response

Dear Editor,

Thank you very much for your helpful comments. We have done our best to address your concerns.

I think that this paper provide a comprehensive review of our experience, with an adequate sample size, in the management of patients with ductal carcinoma in situ (DCIS).

All the data obtained in this study are in line with those reported in the most recent worldwide literature.

We performed a retrospective observational analysis of our experience in the treatment of "pure" DCIS cases (without invasive components) with a large and consecutive series in 9 years period.

Large series are useful to confirm the current standard treatments and to share the daily clinical experience.

Thanking you once more for your useful comments, we sincerely hope that you will find the revisions satisfactory.

Reviewer 3 Report

Chiappa et al. provide a comprehensive review of their experience treating patients with ductal carcinoma in situ (DCIS). It would be of interest to the readers if the author of the manuscript describe any pharmacological interventions utilized in addition to the surgical treatment of DCIS. Do patients require any anticancer therapy following the surgical removal of the lesions? What type of therapy and how long?

Author Response

Dear Editor,

Thank you very much for your helpful comments. We have done our best to address your concerns.

In accordance with national and international guidelines we have offered to our patients adjuvant treatment with radiotherapy ± endocrine therapy according to receptors status and/or the surgical treatment performed (we discuss the adjuvant treatment from line 221 to 231).

The hormonal therapy was chosen based on the patient's menopausal status and the two most used drugs were tamoxifen for pre-menopausal or aromatase inhibitor for post-menopausal patients.

The duration of hormone therapy was 5 years.

A standard therapeutic protocol of radiotherapy with a whole breast irradiation (WBI) typically to a dose of 50 Gy in 25 fractions and a tumor bed boost of 10 Gy in 5 fractions is always for patients undergone conservative surgical treatment.

Thanking you once more for your useful comments, we sincerely hope that you will find the revisions satisfactory.

Reviewer 4 Report

Dear authors,

congratulations for this large and closely monitored database of consecutive pure DCIS of a 9-year period. 

Distribution of different types of DCIS are well described and their treatment in the long time period accordingly. 

The manuscript would benefit of some comments on strategies to cut down the comparably high rate of unclear margins of almost 30 % (please explain the underlying strategy between 2014-2018 to decrease positive margins and the potential interaction with the raise of mastectomy in this period). 

Yours sincerely.

Author Response

Dear Editor,

Thank you very much for your helpful comments. We have done our best to address your concerns.

We have decided to divide the patients into two distinct groups as in our center since 2014 we have introduced some intraoperative strategies to improve the accuracy and precision of surgical resection.

Since 2014 we have routinely performed the intraoperative assessment of the surgical resection margins performed by the pathologist who “in real time” guides the surgeon to a possible widening of the surgical excision margins. Another strategy, in the case of microcalcifications, was to perform an x-ray of the specimen to evaluate their complete excision.

In our Institute since 2014 few surgeons have been dedicated exclusively to Breast Unit which has allowed the development of an ultra-specialization with an improvement in intra and post operative results.

The increase number of mastectomies in the period 2014-2018 is certainly due to the improvement of diagnostic techniques with the introduction of tomosynthesis and the increasing use of MRI which have made it possible to evaluate more accurately the extent of the neoplasia and in many cases resulting in a change of the surgical strategy.

Thanking you once more for your useful comments, we sincerely hope that you will find the revisions satisfactory.